# How Well Does a General Multimodal Foundation Model Understand 3D CT Scans?

**Lukas Buess**[*1] [ID]                                          Lukas.Buess@fau.de
**Franziska Weber**[*1] [ID]                              franziska.fw.weber@fau.de
**Andreas Maier**[1] [ID]                                   Andreas.Maier@fau.de
[1] *Pattern Recognition Lab, Friedrich-Alexander-Universität Erlangen-Nürnberg, Erlangen, GER*

## Abstract

General multimodal foundation models have shown strong performance in natural image and video understanding, but their ability to reason over 3D medical imaging remains underexplored. In this work, we investigate how well Kimi-K2.5, a novel general multimodal model, transfers to volumetric CT interpretation when chest CT scans are converted into video inputs. We evaluate on the CT-RATE validation set, a large-scale benchmark for 3D chest CT understanding, using two tasks: radiology report generation and multiple-choice visual question answering. As a domain-specialized reference, we compare against CT-CHAT. We find that CT-CHAT produces reports that align better with the reporting style reflected in the CT-RATE dataset, whereas both models achieve comparable performance on selected clinically relevant report evaluation metrics. For multiple-choice visual question answering, Kimi-K2.5 achieves meaningful zero-shot performance, although it does not match the performance of CT-CHAT. Overall, our results suggest that general multimodal pretraining can transfer to 3D CT reasoning.

**Keywords:** CT, foundation models, report generation, visual question answering

## 1. Introduction

Large language models and multimodal foundation models are increasingly being explored for medical imaging tasks, including report generation and question answering (Zhang et al., 2025; Liu et al., 2026; Buess et al., 2025; Acosta et al., 2024). However, most general-purpose multimodal models are trained primarily on natural images and videos, while clinical CT data is volumetric and requires 3D spatial reasoning. It therefore remains unclear how well such models transfer to medical volume understanding without domain-specific pretraining.

Recent medical vision-language work has introduced specialized models and benchmarks for CT reasoning (Hamamci et al., 2026, 2024a,b). In parallel, general multimodal models continue to improve rapidly (Team et al., 2026), motivating the question of whether their broad pretraining already transfers to 3D medical image understanding.

In this work, we study this question using Kimi-K2.5[1] (Team et al., 2026), an open-source general multimodal foundation model. Since the model accepts video inputs rather than volumetric CT directly, we convert CT scans into videos, similar to prior work that also represents CT volumes as videos for other downstream tasks (Fang et al., 2026; Buess et al., 2024), and evaluate the resulting pipeline on CT-RATE (Hamamci et al., 2026).

---

[*] Equal Contribution

1. https://huggingface.co/moonshotai/Kimi-K2.5

We evaluate two tasks: radiology report generation and multiple-choice visual question answering (VQA). As a domain-specific reference, we compare against CT-CHAT[2] (Hamamci et al., 2026), a vision-language model finetuned on CT volumes.

## 2. Methods

We evaluate Kimi-K2.5 on the CT-RATE validation set for radiology report generation and multiple-choice VQA, using CT-CHAT as a CT-specialized reference. Report generation is evaluated on 1,564 CT scans (3,039 reconstructions). For question answering, after excluding invalid multiple-choice pairs, we evaluate 2,814 questions.

**CT-to-Video Preprocessing.** Because Kimi-K2.5 accepts video inputs, we convert each CT volume into a video sequence. CT scans are resampled to 1.25 mm isotropic spacing, represented by 256 axial slices, and center-cropped to $336 \times 336$ pixels. For this conversion, we use the RAVE framework (Agrawal et al., 2025)[3].

**Models and Prompts.** We evaluate Kimi-K2.5, a general multimodal mixture-of-experts model with 1T parameters and 32B active parameters, in a zero-shot setting and compare it to CT-CHAT, a CT-specialized 8B model trained on CT-RATE (Team et al., 2026; Hamamci et al., 2026). For report generation, we use a fixed prompt with `FINDINGS` and `IMPRESSION` sections. For question answering, we use a constrained multiple-choice prompt requiring the final answer in the format `ANSWER: (x)`. Prompts are shown in Figure 1.

```
Report Generation Prompt

Generate a radiology report for the given CT.
Use exactly the following format:

FINDINGS:
<findings section here>

IMPRESSION:
<impression section here>
```

```
VQA Prompt

Answer the question based on the CT scan.

Output format requirements:
1) Brief reasoning before the final answer.
2) Use this answer format:  "ANSWER: (x)"
3) x must be one of:  a, b, c, d
4) Don't add text after the final ANSWER line.

Question:  {question}
```

Figure 1: Kimi-K2.5 prompts for report generation (left) and multiple-choice VQA (right).

**Evaluation.** For report generation, we report clinical, natural language generation (NLG), and classification metrics, following the VLM3D challenge[4] and RadEval-based evaluation (Hamamci et al., 2025; Xu et al., 2025). For VQA, we report answer accuracy.

## 3. Results

Table 1 summarizes the CT-RATE results. For report generation, CT-CHAT scores higher on GREEN, RaTE, RadGraph, BLEU, and BERTScore, whereas Kimi-K2.5 matches CRG

---

2. https://huggingface.co/datasets/ibrahimhamamci/CT-RATE/tree/main/models/CT-CHAT

3. RAVE CT-to-Video: https://github.com/YalaLab/rave

4. VLM3D Challenge: https://reportgen.vlm3dchallenge.com

and achieves higher macro precision and F1. This suggests better alignment of CT-CHAT with the reporting style in CT-RATE, while Kimi-K2.5 remains competitive on clinically relevant metrics. For VQA, CT-CHAT attains higher accuracy. Overall, the results show meaningful zero-shot transfer to CT understanding from axial CT-videos.

Table 1: Report generation is evaluated using clinical, NLG, and classification (CL) metrics. VQA is evaluated using accuracy (**bold**: best; ▨ : VLM3D and VQA metrics).

| Method | Report Generation | | | | | | | | | VQA |
|---|---|---|---|---|---|---|---|---|---|---|
| | Clinical ↑ | | | | NLG ↑ | | CL (macro) ↑ | | | |
| | GREEN | RaTE | RadGraph | CRG | BLEU | BERT | P | R | F1 | Acc. ↑ |
| CT-CHAT | **0.437** | **0.664** | **0.200** | **0.367** | **0.203** | **0.611** | 0.354 | **0.158** | 0.169 | **91.2** |
| Kimi-K2.5 | 0.170 | 0.526 | 0.041 | **0.367** | 0.006 | 0.389 | **0.470** | 0.144 | **0.178** | 77.4 |

**Qualitative example.** Figure 2 shows a representative VQA example from Kimi-K2.5. In this case, the model correctly identifies the relevant findings and selects the correct answer. Its response also refers to specific time points in the CT-video, linking them to anatomical regions and observed findings. Although a single example is not sufficient to assess overall performance, it indicates that the model can refer to temporally localized information from the slice sequence even without CT-specific training.

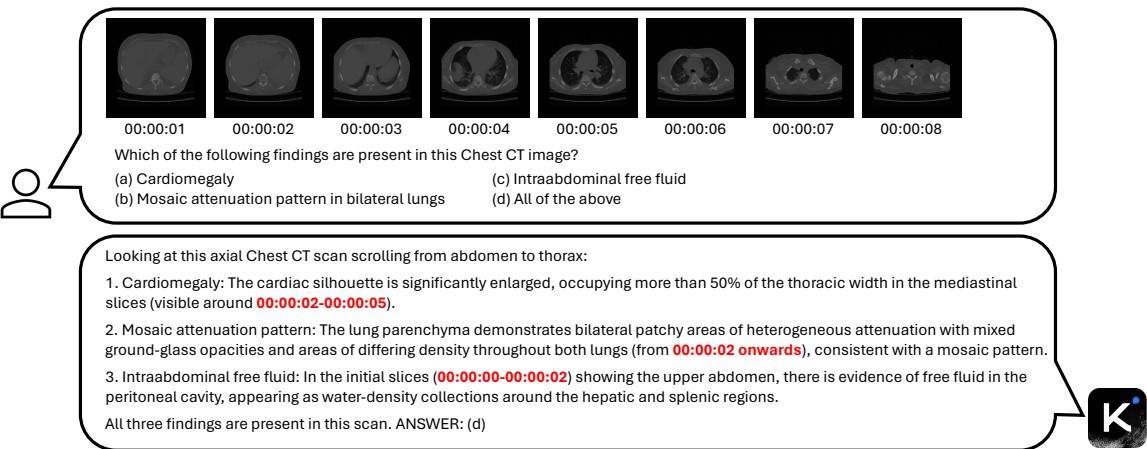

Figure 2: VQA example from Kimi-K2.5 with correct answer and temporal grounding.

## 4. Discussion and Conclusion

Our results show that general multimodal pretraining can transfer to 3D chest CT understanding when CT volumes are represented as videos. Kimi-K2.5 achieves meaningful zero-shot performance on both report generation and VQA. At the same time, domain-specific training improves alignment with the reporting style in CT-RATE and yields better VQA performance, highlighting the continued value of domain-specific training.

## Acknowledgments

The authors gratefully acknowledge the scientific support and HPC resources provided by the Erlangen National High Performance Computing Center (NHR@FAU) of the Friedrich-Alexander-Universität Erlangen-Nürnberg (FAU). The hardware is funded by the German Research Foundation (DFG).

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
