# OpenReview forum: "How Well Does a General Multimodal Foundation Model Understand 3D CT Scans?"
_MIDL.io/2026/Short_Papers — MIDL 2026 - Short Papers Poster_

### Official Review · Reviewer_XVc7 · 2026-05-02

**Rating:** 4
**Confidence:** 4

**Review:**

This paper provides a high-quality assessment of the current boundaries between generalist and specialist AI in medical imaging by leveraging video-based architectures to process volumetric data. The methodology is original and technically sound, utilizing the 1T-parameter Kimi-K2.5 model to achieve impressive zero-shot results that include temporal grounding, where the model links specific findings to time points in the slice sequence. While there remains a notable performance gap in VQA accuracy and reporting style compared to domain-specific models like CT-CHAT, the findings are significant for demonstrating that general-purpose pretraining on natural video contains transferable knowledge for complex 3D medical reasoning.

**Summary:**

The authors evaluate Kimi-K2.5 on the CT-RATE benchmark for two primary tasks: radiology report generation and multiple-choice visual question answering (VQA). Since the model is designed for video rather than 3D volumes, CT scans are resampled and converted into 256-slice axial video sequences using the RAVE framework. Results indicate that while CT-CHAT exhibits superior alignment with the specific reporting style of the dataset, Kimi-K2.5 matches or exceeds its performance on certain macro-classification metrics ($F1$ of $0.178$ vs $0.169$). The model also demonstrates a meaningful zero-shot VQA accuracy of $77.4\%$, compared to the specialized reference of $91.2\%$

**Strengths:**

The study successfully proves that general-purpose pretraining on natural video and text contains transferable knowledge for medical 3D reasoning. The model's ability to maintain competitive clinical macro-precision and F1 scores in a zero-shot setting is particularly impressive and suggests that foundation models could serve as strong base-learners for medical applications.

**Weaknesses:**

The primary weakness is the inherent limitation of "video-style" interpretation, which may miss subtle voxel-level details required for highly specialized diagnoses compared to native 3D architectures. Additionally, the paper does not explore if fine-tuning Kimi-K2.5 would yield a "best-of-both-worlds" result, leaving the ceiling of this approach unknown

**Justification Of Rating:**

The paper provides valuable insights into the transferability of general AI to medical domains. The findings are relevant to the MIDL community as they challenge the necessity of ground-up domain training for all tasks, though the superiority of specialized models like CT-CHAT for specific reporting styles remains clear

---

### Decision · Program_Chairs · 2026-05-08

Accept (Poster)